# Microsurgical Anatomy of the Inferomedial Paraclival Triangle: Contents, Topographical Relationships and Anatomical Variations

**DOI:** 10.3390/brainsci11050596

**Published:** 2021-05-04

**Authors:** Grzegorz Wysiadecki, Maciej Radek, R. Shane Tubbs, Joe Iwanaga, Jerzy Walocha, Piotr Brzeziński, Michał Polguj

**Affiliations:** 1Department of Normal and Clinical Anatomy, Chair of Anatomy and Histology, Medical University of Lodz, 90-752 Łódź, Poland; michal.polguj@umed.lodz.pl; 2Department of Neurosurgery, Spine and Peripheral Nerve Surgery, Medical University of Lodz, University Hospital WAM-CSW, 90-549 Łódź, Poland; maciej.radek@umed.lodz.pl; 3Tulane Center for Clinical Neurosciences, Department of Neurosurgery, Tulane University School of Medicine, New Orleans, LA 70112, USA; shane.tubbs@icloud.com (R.S.T.); iwanagajoeca@gmail.com (J.I.); 4Department of Neurosurgery and Ochsner Neuroscience Institute, Ochsner Health System, New Orleans, LA 70433, USA; 5Tulane Center for Clinical Neurosciences, Department of Neurology, Tulane University School of Medicine, New Orleans, LA 70112, USA; 6Department of Anatomical Sciences, St. George’s University, Grenada, West Indies; 7Department of Surgery, Tulane University School of Medicine, New Orleans, LA 70112, USA; 8Department of Anatomy, Kurume University School of Medicine, 67 Asahi-machi, Kurume, Fukuoka 830-0011, Japan; 9Department of Anatomy, Jagiellonian University Medical College, 33-332 Kraków, Poland; jwalocha@cm-uj.krakow.pl; 10Department of Histology and Embryology, Chair of Anatomy and Histology, Medical University of Lodz, 90-752 Łódź, Poland; piotr.brzezinski@umed.lodz.pl

**Keywords:** abducens nerve, cavernous sinus, dorsal meningeal artery, dura mater/anatomy, microsurgery, skull base/anatomy

## Abstract

The inferomedial triangle is one of the two surgical triangles in the paraclival subregion of the skull base. It is delineated by the posterior clinoid process, the dural entrance of the trochlear nerve and the dural entrance of the abducens nerve. The aim of the present article is to describe the anatomical variations within the inferomedial triangle. Measurements of the triangle’s borders and area were supplemented by detailed observations of the topographical anatomy and various arrangements of its contents. Nine adult cadaveric heads (18 sides) and 28 sagittal head sections were studied. The mean area of the inferomedial triangle was estimated to be 60.7 mm^2^. The mean lengths of its medial, lateral and superior borders were 16.1 mm, 11.9 mm and 10.4 mm, respectively. The dorsal meningeal artery was identified within the inferomedial triangle in 37 out of 46 sides (80.4%). A well-developed petrosphenoidal ligament of Grüber was identified within the triangle on 36 sides (78.3%). Although some structures were variable, the constant contents of the inferomedial triangle were the posterior petroclinoid dural fold, the upper end of the petroclival suture, the gulfar segment of the abducens nerve and the posterior genu of the intracavernous internal carotid artery.

## 1. Introduction

The inferomedial triangle is one of the two surgical triangles in the paraclival subregion of the skull base. According to the classical description by Dolenc [1], it is delineated by the posterior clinoid process (the medial point of the triangle), the dural entrance of the trochlear nerve (the superolateral point) and the dural entrance of the abducens nerve (the inferolateral point). The main contents of this triangle include such anatomical structures as the dura forming the posterior wall of the cavernous sinus, the abducens nerve (typically located in Dorello’s canal under the petrosphenoidal ligament), the petrosphenoidal (Grüber’s) ligament, the posterior genu of the internal carotid artery’s intracavernous segment and the dorsal meningeal artery, which is typically a branch of the meningohypophyseal trunk [1,2,3,4,5,6,7,8]. Rhoton [2] stated that removing the medial half of the triangle exposes the lateral edge of the dorsum sellae and the upper end of the petroclival suture. The inferomedial triangle is adjacent to the inferolateral triangle [9]. It also neighbors the oculomotor triangle located within the middle cranial fossa; the medial half of its upper border contributes to the formation of the oculomotor triangle’s posterior border [1].

There are few detailed descriptions of the normal anatomy of the triangles around the cavernous sinus in the literature, and only the typical arrangement of structures within each triangle is generally shown [1,2,3,4,5,6,7]. A new approach to understanding the triangles around the cavernous sinus addressing the locations and positions of the structures and their three-dimensional relationships was proposed by Chung et al. [10]. However, all these reports were based on the most common anatomical pattern. Therefore, possible anatomical variations within the cavernous sinus’s surgical triangles have not been reported. Precise understanding of the microsurgical anatomy of the paraclival subregion is essential for some advanced neurosurgical procedures. Therefore, the overriding aim of the present work was to describe the anatomical variations within the inferomedial paraclival triangle. Measurements of the triangle’s borders and area were supplemented by detailed observations of the topographical anatomy and various arrangements of its contents. Such observations can extend the scope of previous works and improve our understanding of the microsurgical anatomy of the inferomedial triangle.

## 2. Materials and Methods

The study was conducted according to the guidelines of the Declaration of Helsinki and approved by the bioethics committee of the Medical University of Lodz (protocol code: RNN/518/14KB, with further amendment KE/322/21). Nine adult cadaveric heads (18 sides) and 28 adult sagittal head sections fixed in 10% formalin were studied. A total of 46 sides (23 right and 23 left) were included. There was no evidence of prior trauma or neurosurgical interventions in the material examined. The arteries in one head were injected with Plastogen G (Plasto-Schmidt, Speyer, Germany) stained with red pigment. The first stage of the procedure involved assessing the clival and paraclival dura, and then identifying the inferomedial triangle tips and evaluating the subarachnoid segment of the abducens nerve. The inferomedial triangle’s borders were identified according to the description by Dolenc [1]. The superior edge of the triangle extends between the dural entry point of the trochlear nerve and the tip of the posterior clinoid process. The medial border is marked out between the dural entry point of the abducens nerve and the tip of the posterior clinoid process, while the lateral edge is the line connecting the dural entry points of the abducens and trochlear nerves.

A Digimatic digital caliper (Mitutoyo Company, Kawasaki-shi, Kanagawa, Japan) was used for the measurements. The area of the inferomedial triangle was estimated using Heron’s formula, which calculated the area from measurements of the three sides. IBM SPSS Statistics software was used for statistical analysis. Basic descriptive statistics were used to structure the raw data. Statistically, we analyzed the correlation between the lengths of the inferomedial triangle borders. The Kolmogorov–Smirnov test of normality was applied to check the data distribution. Pearson’s correlation coefficient was used to measure the statistical relationship between the lengths of the inferomedial triangle borders, as well as the direction of the relationship.

The next stage of the procedure involved excision of the dura mater along the infero-medial triangle’s borders. Coagulated blood was removed from inside the cavernous sinus, which allowed the topographical relationships to be visualized. All anatomical structures exposed at this stage were traced. Subsequently, the access was widened by removing the clival and paraclival dura. Preliminary observations were confirmed in this way, especially regarding the presence and course of the dorsal meningeal artery and the presence and location of the petrosphenoidal ligament. Special attention was given to detailed analysis of any anatomical variations within the triangle. To supplement the gross anatomical observations, two specimens of the abducens nerve were examined histologically using the classical paraffin method to reveal the relationships between the nerve’s dural covering and the periosteal dura. These specimens were stained with hematoxylin-eosin (first specimen) or Hansen’s hematoxylin (second specimen). Two specimens with an ossified posterior petroclinoidal dural fold were also subjected to histological evaluation to confirm the presence of bony tissue; Hansen’s hematoxylin stain was applied in those cases to visualize the osteons better. Histological slices were assessed using an OPTA-TECH MB 200 Series biological microscope (OPTA-TECH, Warsaw, Poland) with an OPTA-TECH HDMI CAM microscope camera and HDMI CAM embedded software.

## 3. Results

### 3.1. Inferomedial Triangle Measurements and Area

The mean area of the inferomedial triangle in the specimens examined was 60.7 mm^2^. The mean lengths of the medial, lateral and superior borders were 16.1 mm, 11.9 mm and 10.4 mm, respectively. Basic descriptive statistics (minimum value, maximum value, median and standard deviation) for the measurements and area of the inferomedial triangle are presented in Table 1. Figure 1 is a box plot depicting the lengths of the triangle’s borders.

The data obtained did not differ significantly from normal distributions. The lengths of the medial and lateral borders of the triangle were strongly positively correlated (high positive correlation), where *r*(44) = 0.697. There was also a moderate positive correlation between the triangle’s medial and superior border lengths, where *r*(44) = 0.579. The correlation between the superior and lateral border lengths was also moderate and positive, where *r*(44) = 0.466.

### 3.2. Topography and Anatomical Variations within the Inferomedial Triangle

The topographical relationships within the inferomedial paraclival triangle were complex (Figure 2). The constant contents of the inferomedial triangle were the posterior petroclinoid dural fold, the upper end of the petroclival suture, the gulfar segment of the abducens nerve and the posterior genu of the intracavernous internal carotid artery (Figure 2 and Figure 3). The gulfar segment of the abducens nerve covered by the dural sleeve was located in the lateral half of the triangle. A single abducens nerve trunk was observed in 44 of 46 sides examined (95.7%; see Figure 2, Figure 3 (left side) and Figure 4). However, the abducens nerve was duplicated in two cases (4.3%), in which two dural entry points were observed (Figure 3, right side). In these cases, the inferomedial triangle was replaced by a trapezoid-shaped area (Figure 3B, right side). Histologically confirmed heterotopic ossification of the posterior petroclinoid dural fold was observed in two cases (Figure 5). This ossification involved approximately the upper two thirds of the inferomedial triangle.

The dorsal meningeal artery was identified within the inferomedial triangle in 37 out of 46 sides (80.4%; see Figure 2, Figure 4B,C and Figure 6A). In two of those, only a tiny part of the artery traveled within the triangle on the verge of its medial border. In all 37 cases, the artery was located in the medial half of the triangle. However, in five cases (5/46, 10.9%), it was found outside the triangle (it traveled medially to the triangle’s medial border; Figure 4D). In the remaining four cases (4/46, 8.7%), the dorsal meningeal artery was not found.

A well-developed petrosphenoidal ligament (of Grüber) was identified within the inferomedial triangle in 36 cases (36/46 = 78.3%; see Figure 2 and Figure 4C). In seven cases (7/46, 15.2%), Grüber’s ligament was rudimentary (very thin). In one case (1/46, 2.2%), it was located outside the triangle, just below the abducens nerve’s dural entrance. In that specimen, the lateral border of the dorsum sellae was moved away from the triangle’s medial border. In the remaining two cases (2/46, 4.3%), the ligament was absent (Figure 4B).

### 3.3. Histological Examination

Histologically, a thin piece of the compact bone (compact cortical tissue) separated Dorello’s canal from the spongy cancellous bone (*diploë*) of the petrous apex. The periosteal layer of the dura mater was seen as dense, irregular connective tissue lining the floor of the petroclival venous confluence (Figure 6). The abducens nerve was surrounded by the meningeal cuff. In the first specimen, the aberrant fascicle of the abducens nerve surrounded by meningeal coverage common with the main nerve’s trunk was detected upon histological examination (Figure 6A). In the second specimen, the dural covering (sleeve) of this nerve was fixed to the periosteal dura lining the bottom of Dorello’s canal (Figure 6B). The presence of atypical ossification within the dura mater covering the paraclival area was histologically confirmed for both specimens with this finding. Bony tissue embedded within dense irregular connective tissue was found during histological evaluation (Figure 5C,D).

## 4. Discussion

Only two papers gave the exact dimensions of the triangles of the cavernous sinus. The first was by Watanabe et al. [3] (Table 1), who estimated the mean area of the inferomedial triangle as 45.9 mm^2^. The second, by Isolan et al. [4], reported the mean area in their sample as 41.79 mm^2^. The result in the present study was slightly higher at 60.7 mm^2^. The area and measurements of the inferomedial triangle, including comparisons with data from Watanabe et al. [3] and Isolan et al. [4], are presented in Table 1. Both the shape of the inferomedial triangle and the length of its edges varied among individuals. The medial border was the longest of the three. The correlation between the lengths of the lateral and medial edges was a logical consequence of the fact that the measurements of both edges depended on the location of the abducens nerve’s dural entry point. Comparing our results with those of other authors, we found that differences in the mean length of the inferomedial triangle’s borders were observed mainly for its superior border (see Table 1). This border’s length showed significant interindividual variability in our sample. The maximal length of the superior border in our sample was 15.1 mm (median = 10.2 mm). This finding was probably the main cause of the differences between the triangle’s area reported in our study compared with those of previous reports.

The posterior clinoid process is an important surgical landmark [1,11]. The distance between the abducens nerve dural entrance and the posterior clinoid process is highly variable (see Table 1); it ranged between 11.9 mm and 20.9 mm in this study. According to Iaconetta et al. [12], the nerve reaches the dural porus on average 17.09 mm (SD = 1.51 mm) from the posterior clinoid process. In another report, the distance varied even more, being between 13 mm and 23.1 mm (mean = 19.5 mm) [13]. However, as Youssef and van Loveren [11] stressed, there can be considerable variations in the size of the posterior clinoid process both among individuals and between sides in the same individual. The size and location of the posterior clinoid process also varied in our study and influenced the shape and size of the inferomedial triangle (see the different positions of the posterior clinoid processes in Figure 3).

The border lengths and area of the inferomedial triangle are not the only determinants of the conditions in the surgical field. It is also essential to analyze the topographical relationships, with particular emphasis on anatomical variations. The inferomedial triangle is a space in the posterior wall of the cavernous sinus. Several anatomical structures of surgical importance are located within it [1,2,3,4,5,6,7]. Its superior border involves the posterior petroclinoid dural fold [1,2,3,4]. Occasionally, that fold is ossified or calcified. Such variation was observed in our material in two cases in which the presence of ossification within the dura covering inferomedial triangle was confirmed histologically. Similar findings have also been described in anatomical and radiographic reports [14,15,16,17,18,19]. Touska et al. [18] concluded that ligamentous skull base mineralizations are relatively common findings on CT scans. In the majority of cases, intracranial calcifications or ossifications are not accompanied by any detectable pathology [14,19]. However, mineralization of skull base ligaments can occur as a result of an interplay between a broad range of factors, including genetics, metabolic abnormalities and mechanical stress [18]. Calcification of the posterior petroclinoid dural fold may also be qualified as age-related degeneration, showing a laminar or nodular pattern [19]. Such variants are clinically relevant and can result in barriers to minimally invasive surgical approaches [1,14,15,16,17,18,19]. Posterior petroclinoid fold ossifications or calcifications can affect surgical dissection of the entry corridor through the inferomedial and inferolateral triangles and can be especially important when access involving the inferomedial triangle is enlarged by opening the oculomotor trigone [1]. The posterior petroclinoid dural fold is also a border between the sphenopetrosal venous gulf and the cavernous sinus [12].

One of the main contents of the inferolateral triangle is the gulfar segment of the abducens nerve. According to the original description by Iaconetta et al. ([12], p. 10), “Within the gulf, the nerve runs superiorly and medially (first knee), passing through Dorello’s canal, superolaterally between Grüber’s ligament, medially over the lateral edge of the dorsum sellae, inferiorly by the superior aspect of the clivus, and laterally to the bone of the petrous apex (area of the sphenopetrous synchondrosis).” The upper part of this segment lies just posterior to the intracavernous carotid artery at the level of the floor of the sella turcica and then enters the posterior cavernous sinus [12,20]. As Barges-Coll et al. [20] stressed, the gulfar segment of the abducens nerve is also the most non-mobile segment. This can be explained by fixation of the nerve’s dural sleeve to the periosteal layer of the dura (see Figure 6).

In the context of anatomical variability within the inferolateral paraclival triangle, the duplication of the abducens nerve should also be discussed. Authorities reporting this variant include Nathan et al. (14.5% of the sample) [21], Umansky et al. (in three of 20 specimens, or 15%) [22] and Iaconetta et al. (with a frequency of 8%) [12]. It should be emphasized that duplication of the abducens nerve can involve a subarachnoidal nerve segment; in such variant two different entry points of the nerve can then be observed, as in our study. However, this duplication can also occur distal to the nerve’s dural entry point (after a single nerve trunk enters the clival dura). Thus, during surgical exploration of the inferomedial triangle area, the possibility of an unexpected accessory trunk of the abducens nerve should be considered.

The petrosphenoidal ligament (also known as Grüber’s ligament, the superior sphenopetrosal ligament or the petroclival ligament) is a useful surgical landmark for the abducens nerve [8,12,20,21,22,23,24,25,26,27]. However, the nomenclature for Grüber’s ligament has been inconsistent in reports in the literature, and some authors used the term petroclinoid ligament instead of the petrosphenoidal ligament [1]. A recent study by Iwanaga et al. investigated the ligament’s morphology and found that Grüber’s ligament does not attach to the posterior clinoid process but to the lateral edge of the clivus. Therefore, the term petroclival ligament should be used when describing the petrosphenoidal ligament [26]. In contrast, the petroclinoid folds are folds of the dura mater that extend between the anterior and posterior clinoid processes and the petrosal part of the temporal bone. The petrosphenoidal ligament, when present, is located under the posterior petroclinoid dural fold. Thus, the use of incorrect terminology for the ligament might have led to earlier erroneous reports.

According to Tomio et al. [23], the petrosphenoidal ligament is visible in most adults. However, it can also be hypoplastic (rudimentary), ossified or even absent [6,8,12,24]. In our series, the ligament was hypoplastic in 15.2% of cases and absent in 4.3%. In the series of Iaconetta et al. [12], the petrosphenoidal ligament was hypoplastic in 3% of cases. In a study by Icke and Ozer [25], the ligament was complete in 52% of cases and incomplete (fragmented or hypoplastic) in 38%. In one of the most recent reports, the ligament was absent in 8.3% of cases [26]. Additionally, the structural relationships between the abducens nerve’s dural covering and Dorello’s canal (both are among the contents of the inferomedial triangle) are clinically relevant; a secondary tunnel forming a “tube within a tube” is located within Dorello’s canal and exclusively contains the abducens nerve [27]. Dural and arachnoid membranous protection of the abducens nerve in the petroclival region was also described by Ozveren et al. [28]. These relationships were also observed in the present study and confirmed by histological examination (Figure 6). Anatomical fixation points should be kept in mind during attempts at surgical mobilization of the abducens nerve.

The dorsal meningeal artery typically runs in Dorello’s canal with the abducens nerve. Umansky et al. [22] found it in this position in 80% of cases. Isolan et al. [29] found the dorsal meningeal artery in 91.6% of cases. However, it is highly anatomically variable [30,31]. In the present study, the artery was found outside the inferomedial triangle in 10.9% of case and was absent in 8.7% of them.

Isolan et al. [5] stressed that although the inferomedial and inferolateral paraclival triangles are not viewed through the endonasal transsphenoidal approach, they might be visualized through the lateral extended endonasal approach. Arbolay et al. [32] concluded that the extended endoscopic endonasal approach is a promising, minimally invasive alternative for particular cases with lesions in the sellar, parasellar or clival regions. Kassam et al. [33] noted the usefulness of expanded endoscopic endonasal approaches as potential options for accessing the middle third of the clivus and the region around the petrous part of the internal carotid artery. Detailed anatomical relationships related to the extended endoscopic endonasal approach to the clivus and craniovertebral junction were described by Cavallo et al. [34]. Those authors stressed that opening the paraclival carotid protuberance permits the widening of the surgical corridor laterally. However, during this maneuver, attention must be given to the AN, which passes together with the dorsal meningeal artery just medially to the ICA. Based on our findings, the dorsal meningeal artery can occasionally occupy a more medial position than usual. Such a location can increase the risk of accidental damage to this vessel. According to Oertel et al. [35], the extended endonasal endoscopic approach to the skull base is still under investigation. Thus, data on anatomical variations within the paraclival area will potentially, with time, become more important to the skull base surgeon who uses endoscopic procedures.

### Study Limitation

As most cases in this study were derived from isolated head sections, we could not determine the age or sex. It is also why a reliable correlation of the dimensions of the inferomedial paraclival triangle with anthropometric measurements of the skull could not be conducted on the examined sample. The measurements of the inferomedial triangle should be compared to differences in basic morphometric measurements of the skull in further studies of intact heads. Such research should be conducted on samples with a known age and sex. Differences between the sides of the same specimen (asymmetry) should also be considered in a large sample.

As the meningohypophyseal trunk branching pattern shows considerable anatomical variability, the positions of all branches of this trunk in relation to the triangles around the cavernous sinus should be examined in specimens injected with colored latex or resin. However, our report attempts a detailed analysis of possible deviations from the most typical conditions within the inferolateral triangle and suggests that unexpected anatomical variations can be clinically relevant.

## 5. Conclusions

Numerous anatomical variations are to be expected within the inferomedial paraclival triangle, including duplication of the abducens nerve, hypoplasia or absence of the petrosphenoidal ligament or absence of the dorsal meningeal artery. The constant contents of the inferomedial triangle were the posterior petroclinoid dural fold, the upper end of the petroclival suture, the gulfar segment of the abducens nerve and the posterior genu of the intracavernous internal carotid artery. Possible deviations from the typical arrangement of anatomical structures should always be taken into account during surgical procedures.

## Figures and Tables

**Figure 1 brainsci-11-00596-f001:**
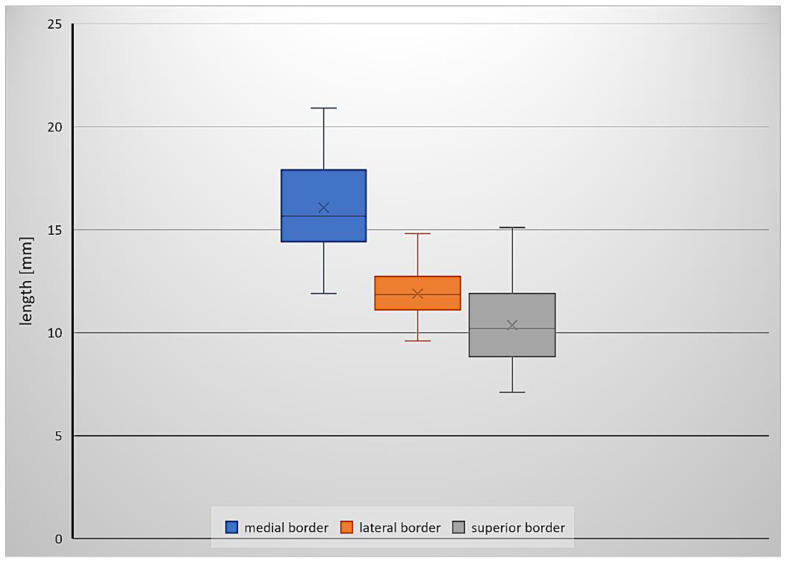
A box plot depicting the border lengths of the inferomedial triangle.

**Figure 2 brainsci-11-00596-f002:**
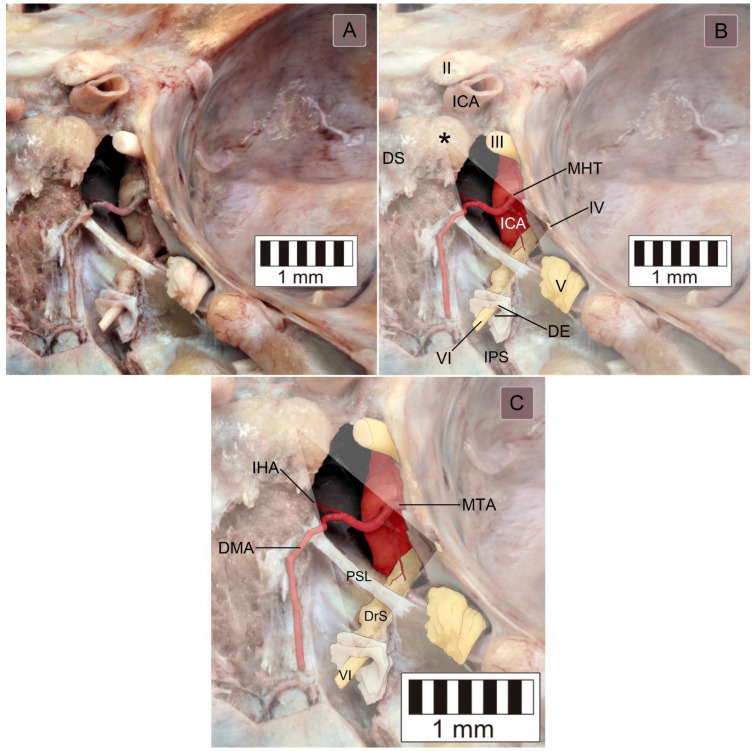
The topographical relationships within the inferomedial paraclival triangle. (**A**) Posterosuperior view of the right paraclival area of a wet anatomical specimen. The arteries were injected with red resin. The petroclival dura, including the posterior wall of the cavernous sinus, was removed. (**B**) The same photograph with neurovascular structures digitally color-enhanced to improve clarity and understanding of the topographical relationships. The borders of the inferomedial paraclival triangle were delineated between the posterior clinoid process (marked by a black asterisk), the dural entry point of the abducens nerve (DE) and the dural entry point of the trochlear nerve (IV). The gulfar segment of the abducens nerve (VI) is located between the dural entry point and the posterior border of the petrosphenoidal ligament. The opening of the inferior petrosal sinus (IPS) is visible below the abducens nerve’s dural entrance. The posterior genu of the intracavernous carotid artery and meningohypophyseal trunk are exposed within the triangle. II = optic nerve; III = oculomotor nerve; V = trigeminal nerve; DS = dorsum sellae; and ICA = internal carotid artery. (**C**) Magnification of (**B**) showing detailed topographical relationships within the inferomedial triangle. The dorsal meningeal artery (DMA) runs posterior and inferomedially. In this specimen, the course of the artery over the petrosphenoidal ligament (PSL) is atypical. The abducens nerve (VI) within the surrounding dural sleeve (DrS) is located at the confluence of the inferior and superior petrosal sinuses with the basilar plexus. IHA = inferior hypophyseal artery and MTA = medial tentorial artery.

**Figure 3 brainsci-11-00596-f003:**
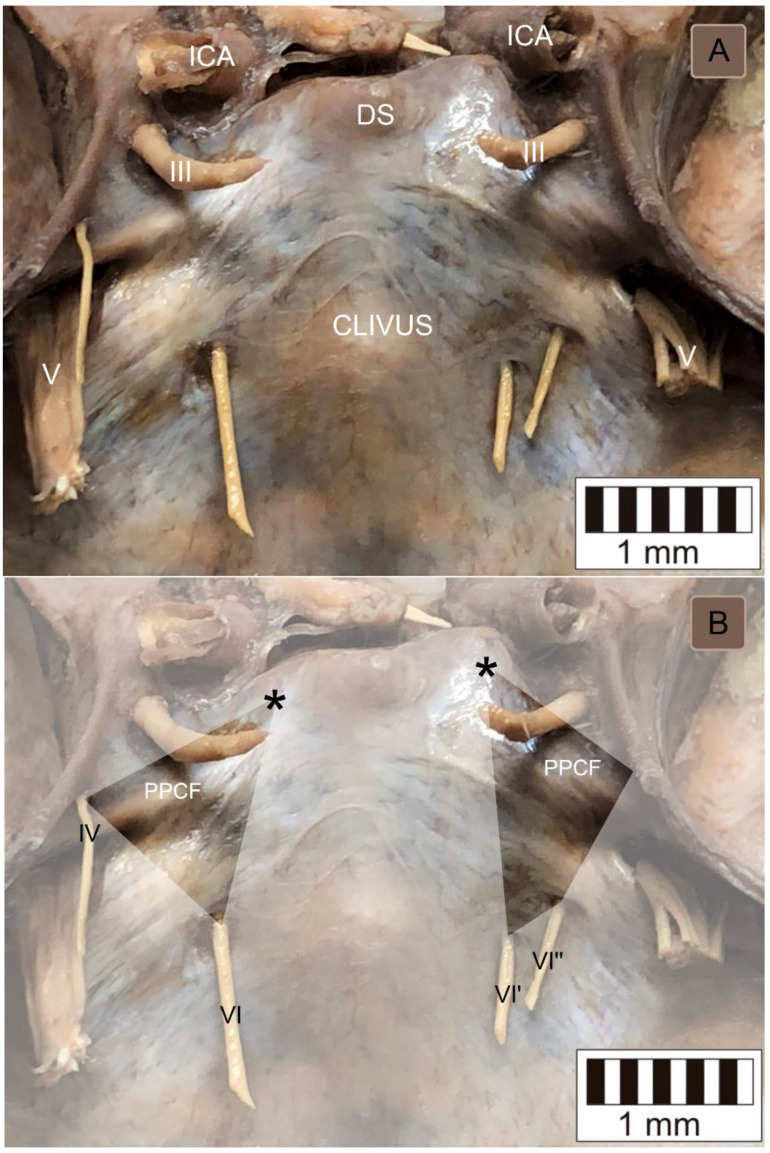
Posterosuperior view to the clivus and paraclival area. (**A**) Wet anatomical specimen. III = oculomotor nerve; V = trigeminal nerve; DS = dorsum sellae; and ICA = internal carotid artery. (**B**) The same photograph with the borders of the inferomedial paraclival triangle marked. On the left side, the anatomical relationships are typical; the triangle is delineated by the single abducens nerve (VI) dural entry point, the dural entrance of the trochlear nerve (IV) and the posterior clinoid process (marked by a black asterisk). Note the different positions (asymmetry) of the posterior clinoid processes on both sides. On the right side, the duplication of the abducens nerve is revealed. In this variant, the two nerve trunks (VI′ and VI″) enter the cavernous sinus posterior wall via two separate dural entry points. On this side, the inferomedial triangle is replaced by a trapezoid-shaped area.

**Figure 4 brainsci-11-00596-f004:**
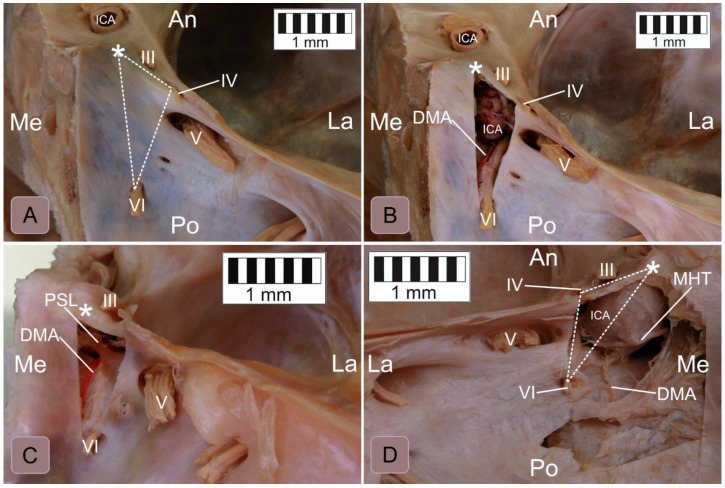
Anatomical variations in the inferomedial paraclival triangle’s contents. (**A**) Posterosuperior view of the right paraclival area. A dashed line marks the inferomedial triangle. (**B**) The same specimen after excision of the dura along the triangle’s borders. The petrosphenoidal ligament is absent. (**C**) Posterosuperior view of the right paraclival area, showing the specimen with the dura excised along the inferomedial triangle’s borders. A well-developed petrosphenoidal ligament (PSL) is exposed. The dorsal meningeal artery (DMA) runs under the ligament within the triangle’s medial half. The abducens nerve (VI), covered by a protective dural sleeve, also runs under the ligament within the triangle’s lateral half. (**D**) Posterosuperior view of the left paraclival area. In this variant, the dorsal meningeal artery travels out of the inferomedial triangle (a dashed line marks the triangle). Note that the posterior genu of the intracavernous carotid artery occupies the upper half of the triangle. The abducens nerve courses along the lateral triangle’s border. In this specimen, the meningohypophyseal trunk (MHT) is located medially to the triangle. III = oculomotor nerve; IV = the dural entrance of the trochlear nerve; V = trigeminal nerve; white asterisk = posterior clinoid process, An = anterior; Po = posterior; Me = medial; and La = lateral.

**Figure 5 brainsci-11-00596-f005:**
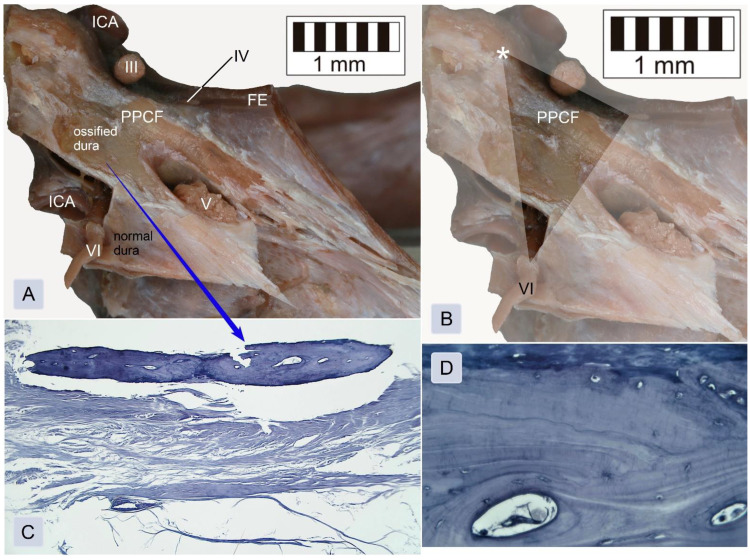
Heterotopic ossification of the posterior petroclinoid dural fold and the dura mater covering the posterior wall of the cavernous sinus seen at the posterior view of an isolated specimen of the right paraclival and parasellar area. (**A**) General view. The external layer of the dura was removed to expose the presence of bony tissue. FE = free edge of the tentorium cerebelli (cut at the petrous apex level); ICA = internal carotid artery; PPCF = posterior petroclinoid dural fold (ossified); III = oculomotor nerve; IV = the dural entrance of the trochlear nerve; V = trigeminal nerve; and VI = abducens nerve. (**B**) The same photograph with marked borders of the inferomedial paraclival triangle. The posterior clinoid process is marked by a white asterisk. (**C**) Histological specimen showing anomalous ossification of the dura mater, covering the upper part of the inferomedial triangle. Heterotopic ossification and dense irregular connective tissue are visualized. The external layer of the dura was removed during the dissection. This histological specimen was stained using Hansen’s hematoxylin and is shown under a 4× objective lens. (**D**) Magnification of (**C**) showing the compact Haversian bone seen under a 40× objective lens. Two osteons with the surrounding lamellae and lacunae of the osteocytes are visualized.

**Figure 6 brainsci-11-00596-f006:**
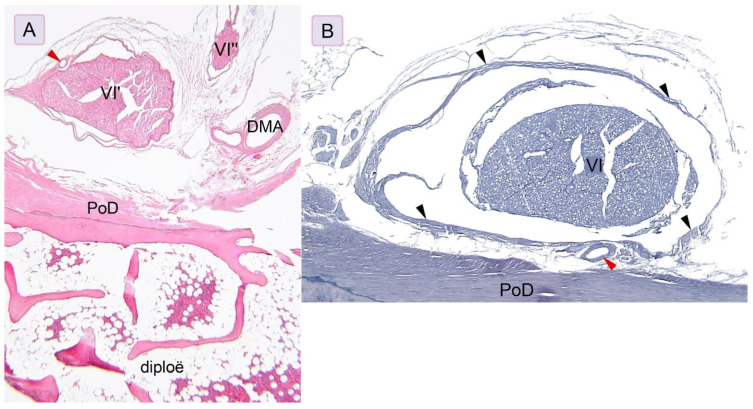
The relationship between the dural sleeve of the gulfar segment of the abducens nerve and the periosteal layer of the dura mater. (**A**) Aberrant fascicle (VI′′) of the abducens nerve (VI′) present at the petrous apex. In this specimen, the dorsal meningeal artery (DMA) is located close to the nerve. In this histological slice, the abducens nerve’s meningeal cuff became detached from the periosteal dura during the histological preparation. Anatomical structures are shown with hematoxylin and eosin staining under a 2× objective lens. (**B**) The abducens nerve’s gulfar segment (VI) is surrounded by the dural sleeve (marked by black arrowheads). The dural covering of the nerve is fixed to the periosteal dura (PoD). Anatomical structures are shown with Hansen’s hematoxylin staining under a 4× objective lens. For both figures, red arrowheads mark small arteries within the dural sleeve.

**Table 1 brainsci-11-00596-t001:** Basic descriptive statistics (minimum value, maximum value, median and standard deviation) for the area and measurements of the inferomedial triangle, including comparisons with data obtained by Watanabe et al. [3] and Isolan et al. [4].

	Medial Border (mm)			Lateral Border (mm)			Superior Border (mm)			Area (mm^2^)		
	This Study	Isolan et al.	Watanabe et al.	This Study	Isolan et al.	Watanabe et al.	This study	Isolan et al.	Watanabe et al.	This Study	Isolan et al.	Watanabe et al.
Min.	11.9	-	-	9.6	-	-	7.1	-	-	33.9	-	-
Max.	20.9	-	-	14.8	-	-	15.1	-	-	96.2	-	-
Mean	16.1	16.22	16.4	11.9	11.37	11.3	10.4	7.25	9.6	60.7	41.79	45.9
Median	15.6	-	-	11.8	-	-	10.2	-	-	60.6	-	-
SD	2.2	0.36	3.4	1.1	0.54	3.4	2	1.03	3.6	15.5	6.45	25

## Data Availability

Data are contained within the article.

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
