# Peer review of "Microsurgical Anatomy of the Inferomedial Paraclival Triangle: Contents, Topographical Relationships and Anatomical Variations"

_brainsci, 2021, doi:10.3390/brainsci11050596_

Round 1

Reviewer 1 Report

This is an interesting anatomical report. Paraclival subregion of the skull base is a complex hidden region. There are only few published reports that provide a detailed anatomy of this this area. In this work, Wysiadecki et al., offer a comprehensive description for the anatomical variations within the inferomedial paraclival triangle which is a part of paraclival subregion. I believe this data is important especially when skull base approaches to reach the upper and middle thirds of the clivus are utilized. Publication of this study appears to be practical from expert opinion

Author Response

This is an interesting anatomical report. Paraclival subregion of the skull base is a complex hidden region. There are only few published reports that provide a detailed anatomy of this this area. In this work, Wysiadecki et al., offer a comprehensive description for the anatomical variations within the inferomedial paraclival triangle which is a part of paraclival subregion. I believe this data is important especially when skull base approaches to reach the upper and middle thirds of the clivus are utilized. Publication of this study appears to be practical from expert opinion.

Response: We thank the reviewer for their thoughtful review of our work and kind words.

Reviewer 2 Report

Dear Authors

The article addresses the important issue of anatomic variations. The topic of the article is not new, but the impressive number of cadaveric and sagittal head sections analyzed provides a strong data set and statistical analysis.

Nevertheless, several concerns should be addressed before considering publication:

Minor concerns:

1. some literature is lacking and should be discussed: eg.

Intracranial Anatomic Triangles: A Comprehensive Illustrated Review.

Doniel Drazin , Joy MH Wang , Fernando Alonso , Daxa M. Patel , Andre Granger ,

Mohammadali M. Shoja , Marios Loukas , Rod J. Oskouian , R. Shane Tubbs

2, Are there enoscopic approaches to the cavernous sinus?

Please add literature. These approaches should also be discussed

3.The diemensions of the inferomedial triangle in your study are different from previous studies and analyses.

In Watanabe et al. and Isolan et al. the results are comparable.

How do you think the different results come about?

4. English grammar and spelling errors should be corrected.

Author Response

The article addresses the important issue of anatomic variations. The topic of the article is not new, but the impressive number of cadaveric and sagittal head sections analyzed provides a strong data set and statistical analysis.

Response: We thank the reviewer for their thoughtful and thorough review.

  1. some literature is lacking and should be discussed: e.g., Intracranial Anatomic Triangles: A Comprehensive Illustrated Review. Doniel Drazin , Joy MH Wang , Fernando Alonso , Daxa M. Patel , Andre Granger, Mohammadali M. Shoja , Marios Loukas, Rod J. Oskouian , R. Shane Tubbs

Response: Thank you for your suggestion. Article of Drazin et al. was cited in text (No. 7 in the references).

  1. Are there enoscopic approaches to the cavernous sinus? Please add literature. These approaches should also be discussed

Response: We added the following paragraph regarding the endoscopic approaches at the end of discussion: “Isolan et al. [5] stress that although the inferomedial and inferolateral paraclival triangles are not viewed through the endonasal transsphenoidal approach, they might be visualized through the lateral extended endonasal approach. Arbolay et al. [32] concluded that the extended endoscopic endonasal approach is a promising minimally invasive alternative for particular cases with lesions in the sellar, parasellar, or clival regions. Kassam et al. [33] noted the usefulness of expanded endoscopic endonasal approaches as potential options for accessing the middle third of the clivus and the region around the petrous part of the internal carotid artery. Detailed anatomical relationships related to the extended endoscopic endonasal approach to the clivus and craniovertebral junction were described by Cavallo et al. [34]. Those authors stress that opening the paraclival carotid protuberance permits the widening of the surgical corridor laterally; However, during this maneuver, attention must be given to the AN, which passes together with the dorsal meningeal artery just medially to the ICA. Based on our findings, the dorsal meningeal artery can occasionally occupy a more medial position than usual. Such location can increase the risk of accidental damage to this vessel. According to Oertel et al. [35], the extended endonasal endoscopic approach to the skull base is still under investigation. Thus, data on anatomical variations within the paraclival area will potentially, with time, become more important to the skull base surgeon who uses endoscopic procedures.”

We cited following new sources:

Arbolay, O.L.; González, J.G.; González, R.H.; Gálvez, Y.H. Extended endoscopic endonasal approach to the skull base. Minim. Invasive Neurosurg. 2009, 52(3), 114-118. doi: 10.1055/s-0028-1119414.

Kassam, A.B.; Gardner, P.; Snyderman, C.; Mintz, A.; Carrau, R. Expanded endonasal approach: fully endoscopic, completely transnasal approach to the middle third of the clivus, petrous bone, middle cranial fossa, and infratemporal fossa. Neurosurg. Focus. 2005, 19(1), E6.

Cavallo, L.M.; Cappabianca, P.; Messina, A.; Esposito, F.; Stella, L.; de Divitiis, E.; Tschabitscher, M. The extended endoscopic endonasal approach to the clivus and cranio-vertebral junction: anatomical study. Childs Nerv. Syst. 2007, 23(6), 665-671. doi: 10.1007/s00381-007-0332-7.

Oertel, J.; Senger, S.; Linsler, S. The extended endoscopic approach to perisellar and skull base lesions: is one nostril enough? Neurosurg. Rev. 2020, 43(6), 1519-1529. doi: 10.1007/s10143-019-01171-8.

  1. The dimensions of the inferomedial triangle in your study are different from previous studies and analyses. In Watanabe et al. and Isolan et al. the results are comparable. How do you think the different results come about?

Response: We thank the reviewer for these observations. We were concerned with these differences. They may result from the fact that the examined sample was heterogeneous and included specimens derived from adult body donors of both sexes. Thus, we decided to extend the study limitations: “As most cases in this study were derived from isolated head sections, we could not determine the age or sex. For this reason, a reliable correlation of the dimensions of the inferomedial paraclival triangle with anthropometric measurements of the skull could not be conducted on the examined sample. The measurements of the inferomedial triangle should be compared to differences in basic morphometric measurements of the skull in further studies of intact heads. Such research should be conducted on samples with a known age and sex. Differences between the sides of the same specimen (asymmetry) should also be considered in a large sample." On the other hand, when comparing our results with other authors, we found that differences in the mean length of the inferomedial triangle borders were observed mainly regarding the superior border (see Table 1). This length showed significant inter-individual variability in our sample. The maximal length of the superior border in our sample was 15.1 mm (median = 10.2 mm). This might be the leading cause of the differences between the triangle’s area reported in our study and previous articles. We added this clarification to the discussion.

  1. English grammar and spelling errors should be corrected.

Response: Thank you for this remark. We have thoroughly re-reviewed the manuscript and corrected any errors we came across (all corrections are marked red in manuscript).